# Optimization of Ultrasound-Assisted Extraction of Yields and Total Methoxyflavone Contents from *Kaempferia parviflora* Rhizomes

**DOI:** 10.3390/molecules27134162

**Published:** 2022-06-29

**Authors:** Wantanwa Krongrawa, Sontaya Limmatvapirat, Supachai Saibua, Chutima Limmatvapirat

**Affiliations:** 1Department of Pharmaceutical Technology, Faculty of Pharmacy, Silpakorn University, Nakhon Pathom 73000, Thailand; song-pt@hotmail.com (W.K.); limmatvapirat_s@su.ac.th (S.L.); 2Pharmaceutical Biopolymer Group (PBiG), Faculty of Pharmacy, Silpakorn University, Nakhon Pathom 73000, Thailand; 3Bangkok Lab & Cosmetic Co., Ltd., 48/1, Moo 5, Nongsaesao Road, Tambon Namphu, Amphoe Muang Ratchaburi, Ratchaburi 70000, Thailand; blc@bangkoklab.co.th; 4Department of Pharmaceutical Chemistry, Faculty of Pharmacy, Silpakorn University, Nakhon Pathom 73000, Thailand

**Keywords:** *Kaempferia parviflora*, ultrasound-assisted extraction, methoxyflavone, Plackett–Burman design, Box–Behnken design

## Abstract

The major bioactive components of *Kaempferia parviflora* (KP) rhizomes, 3,5,7,3′,4′-pentamethoxyflavone (PMF), 5,7-dimethoxyflavone (DMF), and 5,7,4′-trimethoxyflavone (TMF), were chosen as the quantitative and qualitative markers for this plant material. In order to extract bioactive components (total methoxyflavones) from KP rhizomes, ultrasound-assisted extraction (UAE) was proposed as part of this study. Plackett–Burman design (PBD) and Box–Behnken design (BBD) were utilized to optimize the effects of UAE on extraction yields and total methoxyflavone contents in KP rhizomes. First, PBD was utilized to determine the effect of five independent variables on total yields and total methoxyflavone contents. The results indicated that the concentration of the extracting solvent (ethanol), the extraction time, and the ratio of solvent to solid were significant independent terms. Subsequently, BBD with three-level factorial experiments was used to optimize the crucial variables. It was discovered that the concentration of ethanol was the most influential variable on yields and total methoxyflavone contents. Optimum conditions for extraction yield were ethanol concentration (54.24% *v*/*v*), extraction time (25.25 min), and solvent-to-solid ratio (49.63 mL/g), while optimum conditions for total methoxyflavone content were ethanol concentration (95.00% *v*/*v*), extraction time (15.99 min), and solvent-to-solid ratio (50.00 mL/g). The relationship between the experimental and theoretical values was perfect, which proved that the regression models used were correct and that PBD and BBD were used to optimize the conditions in the UAE to obtain the highest yield and total methoxyflavone content in the KP rhizomes.

## 1. Introduction

*Kaempferia parviflora* Wall Ex. Baker (KP) is a non-woody plant native to Thailand belonging to the Zingiberaceae family. Due to the bioactive methoxyflavones found in KP rhizomes, such as 3,5,7,3′,4′-pentamethoxyflavone (PMF), 5,7-dimethoxyflavone (DMF), and 5,7,4′-trimethoxyflavone (TMF), the plant’s rhizomes have frequently been utilized to enhance the health of human systems [1]. These components have different biological effects, including cytotoxicity against ovarian cancer SKOV_3_ and macrophage RAW 264.7 cells, protection of cardiac ischemia–reperfusion injury, enhancement of erectile function, and activation of antioxidant defense [2,3]. In a recent study, we found that KP rhizomes treated with a single dose of 7.5 kGy of ^60^Co gamma radiation had the highest concentration of total methoxyflavones with some biological capabilities [4]. In this work, PMF, DMF, and TMF were selected as bioactive indicators, and the total methoxyflavone content was estimated based on the quantities of these chemicals detected in KP extracts. Validation of analytical methods is a prerequisite for performing chemical evaluations [5]. In order to quantify the total methoxyflavone contents of KP rhizome extracts, it was necessary to develop a validated method for simultaneously determining the concentrations of PMF, DMF, and TMF.

Few studies in the past had addressed the optimum conditions for extracting phenolics and flavonoids from KP rhizomes [6]. However, no research was undertaken to determine the optimum extraction procedures for the total methoxyflavone content of KP rhizomes. Classical extraction techniques, particularly maceration and Soxhlet extraction, are time- and solvent-consuming techniques and may enhance the rate of natural active component degradation during extraction [7,8]. Recently, ultrasound-assisted extraction (UAE) has emerged as a viable method for extracting a vast array of phytochemicals. Multiple factors, such as boosting cell wall breakdown and accelerating the diffusion of chemicals from plant material to extracting solvent [9], have been linked to the high extraction efficiency of UAE based on cavitation events. The UAE was chosen for this study due to its numerous advantages over other traditional extraction techniques, namely its simplicity, shorter extraction time, and lower solvent consumption.

In order to obtain the maximum yield and the highest content of total methoxyflavones, the statistical analysis was required to find out the best condition for UAE. The influences of type and concentration of solvent, time and temperature for extraction, and solvent-to-solid ratio were investigated using Plackett–Burman design (PBD). One of the benefits of screening design was to reduce the non-significant variables from the models. After being screened, the important factors were statistically selected for further process optimization. Box–Behnken design (BBD), a response surface methodology process, has proven to be effective for optimization of extraction. The optimization design provides the optimum conditions with relatively low experimental trials. BBD requires only three levels of each factor and a small number of experimental runs. Furthermore, it does not predict the extreme conditions to obtain the responses [10]. Therefore, BBD was subsequently chosen in this study for process optimization.

As stated previously, the purpose of this study was to identify the most important variables using PBD and then optimize the effect of UAE process variables on the extraction yields and total methoxyflavone contents of KP rhizomes using BBD. In addition, there are no published data that support the evaluation and absolute validation of a spectrophotometric approach for the simultaneous analysis of methoxyflavones in KP rhizome extracts. In terms of developing and fully validating the high-performance liquid chromatography–diode array detection (HPLC-DAD) method for the quantification of PMF, DMF, and TMF, the purpose of the project was to create and implement this method.

## 2. Materials and Methods

### 2.1. Materials and Reagents

The rhizomes were separated from KP plants (Rom-Kloa variety) cultivated in Chat Trakan district, Phitsanulok province, Thailand. The voucher specimen (No. VS001) was deposited in the Faculty of Pharmacy, Silpakorn University, Thailand. KP rhizomes were sun-dried, ground, and separated by sieves with size between 177 and 250 μm before being stored at 25 °C.

PMF, DMF and TMF standards were acquired from Indofine Chemical Company (Hillsborough, NJ, USA). Ethanol (AR grade) and methanol (HPLC grade) were purchased from RCI Labscan (Bangkok, Thailand) and Fisher scientific (Seoul, Korea), respectively. Acetic acid (AR grade) was bought from Merck KGaA (Darmstadt, Germany). Ultrapure water was processed by a TKA Pacific-UP/UPW water purification system (Niederelbert, Germany).

### 2.2. Extraction Process

UAE was performed using a Crest Ultrasonics Powersonic device (230D, Crest Ultrasonics Corporation, Ewing Township, NJ, USA) operating at 40 kHz (frequency) and 160 W (power). The working frequency and power were derived from the preliminary experiments. Five grams of KP rhizome powder was taken into a screw-cap tube and then performed according to the BBD design matrix. A Whatman filter paper No.1 was employed to filter the extract solution. The filtrate was dried using a rotary evaporator (Rotavapor^®^ R-100, Buchi, Japan) at 40 °C. The obtained extract was pre-frozen at −20 °C prior to the freeze-drying process using a freeze dryer (Model 6112974, Labconco, Kansas, MO, USA) for 24 h in order to remove residual water. All dried extracts were kept in the dark at −20 °C.

### 2.3. Instrumentation and Chromatographic Conditions

HPLC-DAD (Agilent 1100 series, Agilent Technologies, Santa Clara, CA, USA) was utilized for investigation of PMF, DMF, and TMF. Chromatographic segregation was performed using a porous spherical silica, Reprosil-Pur Basic C_18_ column (4.6 × 250 mm, 5 µm) obtained from Dr. Maisch HPLC GmbH (Ammerbuch-Entringen, Germany), controlling column temperature at 55 °C. The mobile phase employed was an isocratic elution system of methanol: 0.1% *v*/*v* aqueous acetic acid (70:30, *v*/*v*). The elution was accomplished at a flow rate of 0.8 mL/min, the injection volume was 20 µL, and the UV absorption of each compound was accessed at 254 nm.

### 2.4. Preparation of Standard and Sample Solutions

The standard stock solutions (0.2 mg/mL) were prepared by solubilizing 20 mg of PMF, DMF, and TMF in methanol separately. The standard solutions of marker components were obtained by diluting each standard stock solution with methanol to attain concentrations of 2.5, 5, 10, 20, 40, 60, and 100 µg/mL. Using three sets of standard solutions, calibration curves were generated.

Individually dissolving KP extracts in a mobile phase and adjusting the volume with the same solvent provided 1 mg/mL sample solutions. Each solution was filtered using a 0.45 μm nylon filter.

### 2.5. Method Validation

The developed method was validated with reference to linearity, specificity, precision, accuracy, robustness, limit of detection (LOD), and limit of quantitation (LOQ) in line with the international conference on harmonization (ICH) Q2 (R1) guidelines [11].

#### 2.5.1. System Suitability

Before beginning the experiments, the suitability of the chromatographic systems used for analysis had to be determined. The standard mixture solution of PMF, DMF, and TMF with respective concentrations of 20.80, 22.10, and 25.74 μg/mL was created by diluting the mixture of standard stock solutions with methanol. After stabilizing the HPLC system for 30 min, a blank solution (single injection) and a standard mixture solution (five replicates) were introduced to evaluate the system suitability parameters including retention time, peak area, tailing factor, resolution, and theoretical plate.

#### 2.5.2. Linearity

In order to examine the linearity of the three components, standard concentrated solutions of PMF, DMF, and TMF were diluted serially in methanol to yield seven distinct concentrations ranging from 2.61 to 104.50 μg/mL, 2.76 to 110.51 μg/mL, and 3.22 to 128.74 μg/mL, respectively. The linearity was determined by calculating the regression line from the graph of peak areas versus concentrations of standard solutions using the linear least squares method.

#### 2.5.3. Specificity

Specificity is the capacity to evaluate unambiguously the presence of sample constituents that can be reasonably anticipated to be present. Typically, these include contaminants, sample matrices, and degradation products. In this study, it is determined by comparing the HPLC elution profiles of blank, standard reference, and sample.

#### 2.5.4. Precision

The repeatability (intra-day) and intermediate precision (inter-day) of the method were evaluated to determine its precision. Five repeated analyses of the same standard solution of PMF, DMF, and TMF at seven different concentrations within concentration ranges of 2.61 to 104.50 μg/mL, 2.76 to 110.51 μg/mL, and 3.22 to 128.74 μg/mL, respectively, were performed on the same day and under the same experimental condition to determine repeatability. By analyzing a series of standard solutions for three consecutive days in the same laboratory, the intermediate precision was determined. Data precision was calculated as the percent relative standard deviation (% RSD) using the following formula: % RSD = (SD/mean) 100%.

#### 2.5.5. Accuracy

To assess precision, recovery studies were conducted at 75–125% of the target concentration levels in the KP extract. The spiked samples containing PMF, DMF, and TMF at concentrations of 20, 40, and 60 μg/mL, respectively, were generated by dissolving standard mixtures into the sample solutions (1.27 mg/mL) using methanol as the diluent, and then analyzing them five times. Recovery (%) = [(observed amount − original amount)/spiked amount] × 100.

#### 2.5.6. Robustness

Robustness was verified by scrutinizing five replicates of the same standard solutions of PMF, DMF, and TMF under slightly altered conditions. Two different mobile phase ratios (methanol: 0.1% *v*/*v* aqueous acetic acid, 65:35 and 70:30, *v*/*v*) and detection wavelengths (254 and 254 nm) were employed to determine the method’s robustness.

#### 2.5.7. LOD and LOQ

Signal-to-noise ratio was used to determine LODs and LOQs of PMF, DMF, and TMF. In order to determine these values, standard solutions of methoxyflavones were diluted with the mobile phase until just minor peaks were visible. The LOD was evaluated at a signal-to-noise ratio of 3:1, whereas the LOQ was estimated at a 10:1 ratio.

### 2.6. Determination of Methoxyflavones Using a HPLC-DAD Method

About 50 mg of sample was weighed, and it was dissolved by sonicating it with mobile phase. To achieve a final concentration of 1 mg/mL, the aliquots were diluted with mobile phase. The sample solution was analyzed in triplicate using the validated HPLC-DAD method after testing the system suitability to produce a constant baseline. Regression equations were used to calculate the contents of PMF, DMF, and TMF in the samples.

### 2.7. Experimental Design

The optimization of yields and total methoxyflavone contents from KP rhizomes was split into two parts in this study. First, PBD was used to study the critical variables that affect the extraction process, and then, the significantly selected variables were further improved using the BBD approach.

#### 2.7.1. Plackett–Burman Design (PBD)

PBD was utilized to identify the significant variables among a large number of variables with relatively low experimental runs. In this research, PBD was applied to assess the impacts of various significant process variables on extraction yields, mainly type of organic solvent, solvent concentration, extraction time, extraction temperature, and solvent-to-solid ratio. The 12 experimental runs were created and carried out based on the PBD matrix. The independent variables were assigned as x_1_–x_5_ and evaluated at high (+) and low (−) levels. Prior to conducting the PBD experiments, the high and low levels of each variable were picked out with respect to the preliminary studies. The selected ranges for all variables were considered according to the instrument limitations. According to the results of preliminary studies, solvents with differing polarities could influence both responses. Several studies used methanol and ethanol to extract methoxyflavones [12,13]. In order to determine the extraction yields and total methoxyflavone contents, both methanol and ethanol were used as extraction solvents. Based on our preliminary research, a low percentage of alcohol resulted in a high extraction yield, while a high percentage of alcohol indicated a high total methoxyflavone content. In addition, a longer extraction period increased the extraction yield and total methoxyflavone content. However, prolonged sonication might well generate free radicals from water, resulting in the decomposition of active polyphenols especially methoxyflavones [14]. Consequently, the extraction duration should be chosen within the appropriate range. During ultrasound extraction, the extraction temperature was strongly correlated with extraction time, indicating that an increase in extraction time led to a rise in extraction temperature. It was determined that the maximum extraction time should not exceed 30 min. The ratio of solvent to solid was also considered. Both extraction yield and methoxyflavone content increased as the ratio of solvent to solid increased, according to the findings of preliminary research. At 50 mL/g, the highest extraction yield and total methoxyflavone content were observed; above this ratio, the values of both responses remained constant. For the low and high levels, the range of solvent-to-solid ratios between 10 and 50 mL/g was chosen (Table 1).

Due to the results of preliminary studies, a two-level PBD was chosen to screen the model’s crucial parameters. Although not all interactions between the factors were considered, this design was effective for detecting main effects with relatively few experimental runs (5 factors with 12 runs versus 32 total runs for standard two-level factorial design, 2^k^). For each variable, the low and high levels of solvent concentration (50% *v*/*v* and 95% *v*/*v*), extraction time (5 min and 30 min), extraction temperature (30 °C and 80 °C), and solvent-to-solid ratio (10 mL/g and 50 mL/g) were tested. Additionally, solvent type including methanol and ethanol classified as a categorical variable was also investigated at two levels of low and high, respectively. Table 1 represents the experimental combinations with selected independent variables obtained from PBD. The extraction yield and total methoxyflavone content were taken as the responses. All experimental trials were carried out in triplicate, and the mean values were analyzed by employing Design Expert Software version 8.0 (Stat-Ease, Inc., Minneapolis, MN, USA). The results obtained from PBD were generally expressed in accordance with the first-order polynomial model as shown in Equation (1):Y = β_0_ + β_i_x_i_(1)where Y is the predicted response; β_0_ is the intercept; β_i_ is the coefficient of linear regression; and x_i_ is the level of the independent variable. However, the interaction term for this design is insignificant.

#### 2.7.2. Box–Behnken Design (BBD)

The three most important independent variables, ethanol concentration (x_1_), solvent-to-solid ratio (x_2_), and extraction time (x_3_), affecting extraction yields and total methoxyflavone contents were then investigated by BBD based on the results of PBD experiments. As shown in Table 2, the independent variables were examined at three distinct levels: low (−1), center (0), and high (+1). All experiments were conducted in triplicate, and Design Expert software (version 8.0) was used to analyze the results. Utilizing a polynomial model (Equation (2)), the mathematical relationship between independent variables and response values was determined.
(2)$$Y=\beta_{0}+\sum_{i=1}^{3}\beta_{i}x_{i}+\sum_{i=1}^{3}\beta_{i}x_{i}^{2}+\sum_{i=1}^{3}\beta_{\mathrm{ij}}x_{i}x_{j}$$where Y is the predicted response; β_0_ is the intercept; β_i_ and β_ij_ are the regression coefficients of linear and interactive terms, respectively; x_i_ and x_j_ represent the independent variables.

### 2.8. Data Analysis

The data analysis was carried out using Microsoft Excel and Analysis of Variance (ANOVA, using Design-Expert software (Trial version 8.0.7.1, Stat-Ease Inc., Minneapolis, MN, USA)). The mean and standard deviation of the triplicate values are reported.

## 3. Results and Discussions

### 3.1. Method Development

The developed RP-HPLC method was validated and applied to identify and quantify PMF, DMF, and TMF, which were determined to be marker constituents of all KP rhizome extracts. A suitable HPLC system condition for the separation of methoxyflavones consisted of an RP-C18 chromatographic column (4.6 × 250 mm, 5 μm), column temperature at 55 °C, and mobile phase containing methanol: 0.1% *v*/*v* aqueous acetic acid in the ratio of 70:30 *v*/*v*, at a flow rate of 0.8 mL/min with λ_max_ at 254 nm. Under the aforementioned HPLC conditions, the respective compounds (PMF, DMF, and TMF) were clearly separated and eluted at retention times of 16 min, 19 min, and 21 min, respectively, as illustrated in Figure 1. The developed method revealed sharp symmetrical peaks, whereas the earlier study revealed broad asymmetries [15].

For the developed HPLC-DAD method, a highly accurate and precise separation of marker components was achieved. In addition, no sample pretreatment was required prior to injection into HPLC. The RSD of retention time was between 0.68% and 1.33%, and that of the peak area was between 0.99% and 1.35%, according to the study of system suitability. The tailing factor (0.94–0.95), peak asymmetry factor (0.98–0.99), resolution (2.52–32.28), and number of theoretical plates (11,877–15,631) were in accordance with ICH guidelines [11] and United States Pharmacopeia-National Formulary (USP 43-NF 38) regulation [16]. These outcomes demonstrated that the chromatographic system was suitable for the simultaneous determination of PMF, DMF, and TMF.

In addition, the chromatogram of methanol extract was presented as shown in Figure 1. Under validated conditions, the three analyzed methoxyflavones (PMF, DMF, and TMF) in ethanol extract and methanol extract were completely separated on chromatograms with similar retention times. The previous report also found small differences in a few peaks of the extract when different solvents were used for extraction [17]. In this study, the PMF:DMF:TMF ratios for 95.0% *v*/*v* ethanol extract and 95.0% *v*/*v* methanol extract were approximately 2:2:1.

### 3.2. Validation of HPLC-DAD

The HPLC-DAD method was validated with regard to linearity, specificity, precision, accuracy, robustness, LOD, and LOQ. The assay was demonstrated to be of good linearity, high accuracy, and precision for the three marker components (Table 3 and Table 4). The linear regression equations of PMF, DMF, and TMF exhibited good linear relationships with correlation coefficient (R^2^) values greater than 0.9995. The LODs and LOQs for the three methoxyflavones were established to be in the range of 0.052–1.44 μg/mL and 1.48–4.40 μg/mL, respectively (Table 3), which demonstrated the good sensitivity of the method. The method specificity is illustrated in Figure 1C,E where the absolute separation of three methoxyflavones was observed in the spiked *K. parviflora* extract containing PMF, DMF, and TMF. There was no interference at the retention times of PMF, DMF, and TMF in the chromatogram of the extract. Additionally, the well-shaped peaks also indicate the specificity of the method. As shown in Table 4, the high percent recoveries ranged from 109.00 ± 0.40% to 109.86 ± 3.72% for PMF, 92.23 ± 0.61% to 113.48 ± 0.51% for DMF, and 92.41 ± 0.50% to 100.75 ± 0.88% for TMF. Intra-day and inter-day precision for PMF, DMF, and TMF at a concentration of 2.61–128.74 μg/mL ranged from 0.03% to 0.95% RSD and from 0.34% to 3.15% RSD, respectively (Table 4). The results indicated that the developed method possessed a high degree of accuracy and precision. Furthermore, there were no significant alterations in the chromatographic patterns when the small changes in mobile phase ratio and detection wavelength were performed in the experimental conditions. All RSDs for replicate injections were less than 2.0%. Therefore, the developed method was robust. All the assay results gained from the method validation were consistent with the acceptance criteria [11,18]. In conclusion, the developed HPLC-DAD method was found to be acceptable for determining PMF, DMF, and TMF in KP extracts obtained from the UAE optimization without the need for sample preparation.

### 3.3. Screening of Significant Variables Using Plackett–Burman Design

PBD, a two-level factorial design, was used in this study to assess the influences of independent variables on extraction yields and total methoxyflavone contents as well as to screen the crucial variables for further optimization process. The design matrix of PBD along with the responses are depicted in Table 5.

To evaluate the impact of independent factors on extraction yields and total methoxyflavone concentrations, a Pareto chart, a useful tool for illustrating the relative relevance of response variables, was utilized. The comparison between computed and tabular *t*-values represents the degree effect of each variable, showing that the variable with a *t*-value over the *t*-limit line was the significant variable. Moreover, if the *t*-value is greater than the Bonferroni limit line, the variable is statistically significant. According to Pareto charts (Figure 2), the solvent concentration has the greatest influence on extraction yields and total methoxyflavone contents, which is followed by the extraction duration and solvent-to-solid ratio, which had a modest effect on extraction yields. Solvent type and extraction temperature were eliminated from further study as insignificant variables.

The ANOVA results are shown in Table 6. Greater than 95% confidence (*p* < 0.05) indicates the significance of the variable. R^2^ values for extraction yields (0.9194), and total methoxyflavone concentrations (0.9254) were consistent. According to the results of ANOVA, the concentration of ethanol, the extraction time, and the solvent-to-solid ratio were determined to be crucial variables for KP rhizome extraction yields. In the meantime, ethanol concentration and extraction time were considered to be significant parameters for total methoxyflavone contents. In the subsequent experiments of the present study, ethanol was selected, despite the fact that methanol produced a higher yield and methoxyflavone concentration. This is due to the insignificance of the solvent type impact (*p* > 0.05). In addition, ethanol is indisputably safer than methanol. The extraction temperature variable, the other insignificant variable, was set to 50 °C.

### 3.4. Optimization of Significant Variables by Box–Behnken Design

According to the PBD analysis, the most influential variables were ethanol concentration (x_2_), solvent-to-solid ratio (x_3_), and extraction time (x_5_). BBD then investigated the effects of three independent variables on extraction yields and total methoxyflavone levels. The matrix of BBD and the results of 17 experimental runs are displayed in Table 7. Version 8.0 of Design-Expert was utilized to analyze the experimental design and calculate the anticipated data. According to Table 7, extraction yields (Y_1_) and total methoxyflavone concentrations (Y_2_) ranged from 2.68% to 15.36% and 121.80 to 321.05 mg/g dried extract, respectively. The correlation between the three variables and responses (Y_1_ and Y_2_) was evaluated using multiple regression analysis, and the second-order polynomial equations (Equations (3) and (4)) are depicted as follows:Y_1_ = 9.01 − 4.82x_2_ + 1.34x_3_ + 2.03x_5_(3)
Y_2_ = 166.07 + 89.98x_2_ + 43.55x_2_^2^ + 16.64x_3_^2^(4)

Table 8 displays the findings of an ANOVA. The model F-value of 50.01 indicated that the model was significant at *p* < 0.0001, whereas the lack of fit F-value of 6.47 indicated no significance at *p* = 0.0566. The model F-value of 139.70 indicated that the model was statistically significant at *p* < 0.0001, whereas the lack of fit F-value of 4.76 was not statistically significant at *p* = 0.0743. Both the model’s low *p*-value and the lack of fit term’s high *p*-value indicate that the model is significant. However, an R^2^ value close to 1.0 indicates a nearly perfect positive correlation, and the quadratic model appears highly plausible. In addition, a high R^2^ value indicated a strong correlation between the experimental and predicted responses. From Equations (3) and (4), the predicted R^2^ values of 0.9203 and 0.9790, respectively, were in reasonable agreement with the adjusted R^2^ values of 0.9019 and 0.9720, indicating that both models adequately represent the parameter variability.

Additionally, the other parameters used to verify the validity of the models are required. The adequate (adeq.) precision indicating the signal-to-noise ratio should be greater than 4. As shown in Table 8, the adeq. precision values of 22.65 and 31.50 indicated which models were acceptable and dependable. The model’s reproducibility is also a crucial criterion for determining its practicability. The coefficient of variation (CV) is used to estimate the model’s reproducibility. In general, % CV equal to or less than 10 is regarded as acceptable [19]. The results revealed that percentages of CV for two prediction models were 9.85% and 5.93% (Table 8), implying that the models were reproducible. According to the information presented above, the statistical models developed in this study were shown to be effective at predicting responses.

As shown in Figure 3, ethanol concentration had the greatest effect on extraction yields, which was followed by extraction time and solvent-to-solid ratio. The extraction yields increased significantly as ethanol concentrations decreased (Figure 3A,B). As solvent-to-solid ratio and extraction time increased to 50 mL/g (Figure 3A,C) and 30 min (Figure 3B,C), respectively, a substantial increase in yield was observed. Design Expert software ascertained that the optimum conditions for the highest extraction yield were ethanol concentration of 54.24% *v*/*v*, extraction time of 25.25 min, and solvent-to-solid ratio of 49.63 mL/g.

The extraction yield was high in a highly polar solvent (low ethanol concentration), suggesting that most polar compounds might be extracted using the “like dissolves like” concept [20]. The longer the extraction duration, the more often the solvent molecule would come into encounter with the powdered plant material, resulting in a higher extraction yield. High solvent-to-solid ratio also increased extraction yield, indicating that a larger volume of solvent could enhance the interfacial area between solvent and plant material. The findings of this investigation were consistent with the optimization of the extraction of curcumin from *Curcuma longa* rhizomes [21].

The correlation between total methoxyflavone contents and three independent factors is depicted in Figure 4. Total methoxyflavone levels increased considerably as ethanol concentration increased (Figure 4A,B). It was discovered that the bioactive methoxyflavone markers of KP rhizomes, PMF, DMF, and PMF, possessed a significant affinity for ethanol due to intermolecular forces [22]. In the interim, the total methoxyflavone concentration increased marginally with increasing extraction time, reaching a maximum at approximately 16.00 min (Figure 4A,C). However, neither the extraction period nor the solvent-to-solid ratio had a significant effect on the overall methoxyflavone content. The Design Expert program determined the optimum parameters for the greatest content of total methoxyflavone as follows: ethanol concentration of 95% *v*/*v*; solvent-to-solid ratio of 50 mL/g; and extraction time of 15.99 min.

### 3.5. Verification of Predictive Model

The maximum extraction yield was evaluated under the following conditions: x_2_, 54.24% *v*/*v*; x_3_, 25.25 min; and x_5_, 49.63 mL/g. The experimental data verification is presented in Table 9. The experimental yield was consistent with the predicted yield (16.95%) under optimal conditions (16.95%). In the meantime, the optimum conditions for maximizing the total methoxyflavone content were as follows: x_2_, 95% *v*/*v*; x_3_, 15.99 min; and x_5_, 50.00 mL/g. The experimental total methoxyflavone content was close to the value predicted (327.25 mg/g of extract). The results demonstrated that both models developed for this investigation were precise and reliable.

## 4. Conclusions

This study utilized a two-level PBD to determine the impact of independent variables on yields and total methoxyflavone concentrations. Both responses were significantly affected by ethanol concentration, extraction time, and the ratio of solvent to solid, but solvent type and extraction temperature were not. Then, to improve the critical process parameters, a three-level BBD was implemented and studied for the subsequent process. In addition, response surface analyses were performed to illustrate the effects of extraction conditions on yields and total methoxyflavone contents. In order to maximize the responses, two regression models were developed and then validated.

The optimum conditions for maximizing extraction yield were an ethanol concentration of 54.24% *v*/*v*, an extraction time of 25.25 min, and a solvent-to-solid ratio of 49.63 mL/g; under these conditions, the total yield was 16.95%. For total methoxyflavone contents, the optimum conditions to obtain the highest total methoxyflavone content were as follows: 95.00% *v*/*v* ethanol concentration; extraction time of 15.99 min; solvent-to-solid ratio of 50 mL/g. These conditions produced a total methoxyflavone content of 327.25 mg/g. In conclusion, the experimental values of both responses closely matched the predicted values. Consequently, these results demonstrate that the obtained models were reliable and reproducible.

Combinations of screening designs followed by an optimization procedure could produce more effective results with fewer experiments. Moreover, the UAE is a viable method for the production of KP extracts rich in bioactive methoxyflavones.

## Figures and Tables

**Figure 1 molecules-27-04162-f001:**
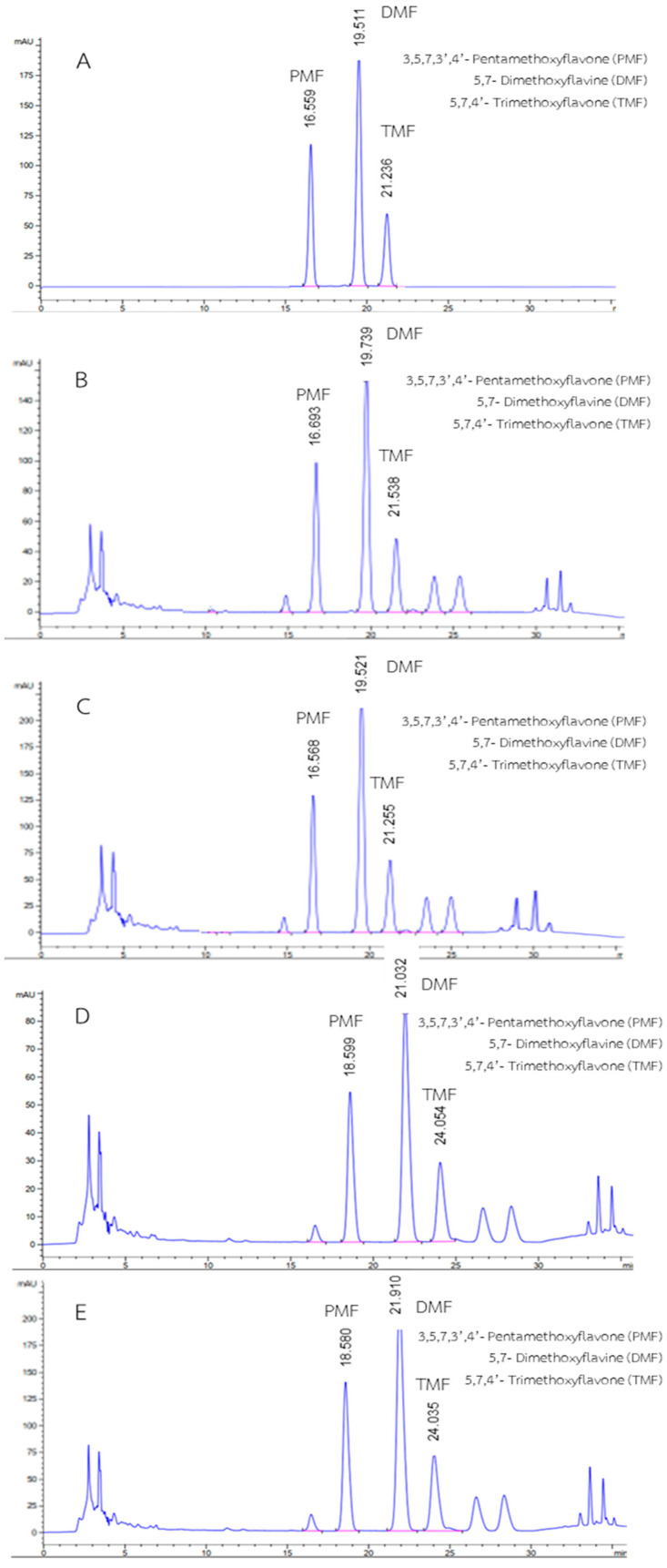
HPLC chromatograms of (**A**) PMF (20.10 µg/mL), DMF (20.05 µg/mL), and TMF (20.12 µg/mL) standards, (**B**) KP rhizome ethanol extract (1.27 mg/mL), (**C**) spiked KP rhizome ethanol extract containing PMF, DMF, and TMF (10 µg/mL), (**D**) KP rhizome methanol extract (1.05 mg/mL), and (**E**) spiked KP rhizome methanol extract containing PMF, DMF, and TMF (10 µg/mL) using UV detection at 254 nm.

**Figure 2 molecules-27-04162-f002:**
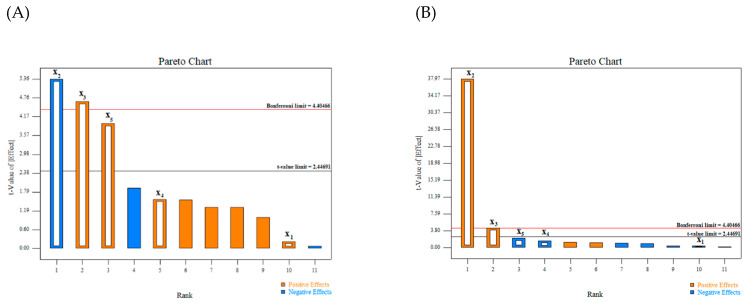
Pareto charts of optimization for dependent variables on the extraction yields (**A**) and total methoxyflavone contents (**B**).

**Figure 3 molecules-27-04162-f003:**
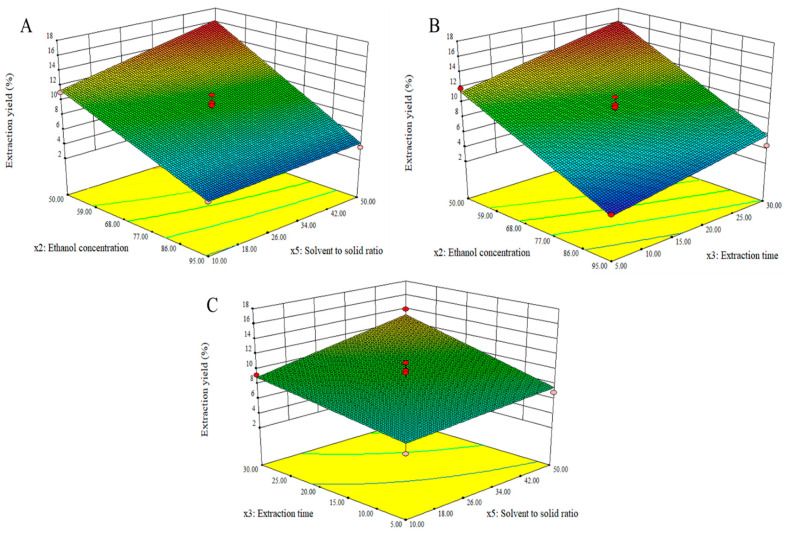
Three-dimensional response surface graphs representing the effect of process conditions on the yields.

**Figure 4 molecules-27-04162-f004:**
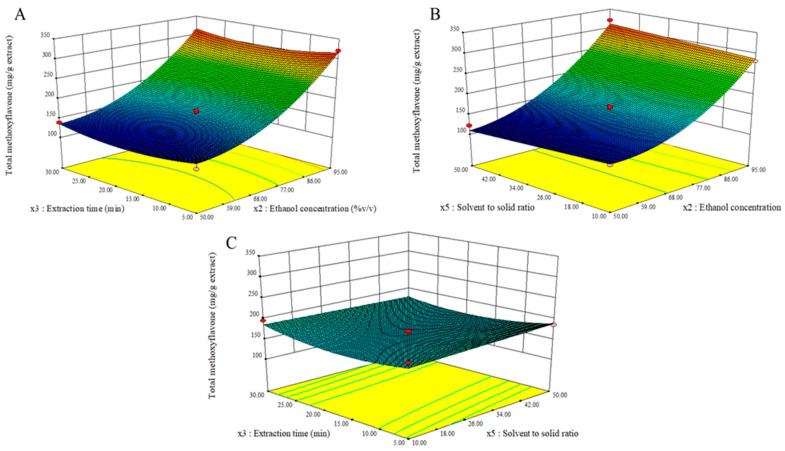
Three-dimensional response surface graphs representing the effect of process conditions on the total methoxyflavone contents.

**Table 1 molecules-27-04162-t001:** Independent variables and corresponding levels tested in PBD.

Symbols	Independent Variables	Levels	
		−1 (Low Level)	+1 (High Level)
**x_1_**	Type of solvent	Methanol	Ethanol
**x_2_**	Organic solvent concentration (% *v/v*)	50	95
**x_3_**	Extraction time (min)	5	30
**x_4_**	Extraction temperature (°C)	30	80
**x_5_**	Solvent-to-solid ratio (mL/g)	10	50

**Table 2 molecules-27-04162-t002:** Independent variables and corresponding levels tested in BBD.

Symbols	Independent Variables	Levels		
		−1 (Low Level)	0 (Center Level)	+1 (High Level)
x_1_	Ethanol concentration (% *v*/*v*)	50	72.5	95
x_2_	Solvent-to-solid ratio (mL/g)	10	30	50
x_3_	Extraction time (min)	5	17.5	30

**Table 3 molecules-27-04162-t003:** Linearity, correlation coefficient (r^2^), limit of detection (LOD), and limit of quantitation (LOQ).

Methoxyflavones	Linearity Range (μg/mL)	Regression Equation	R^2^, *n* = 5	LOD (μg/mL)	LOQ (μg/mL)
PMF	2.61–104.50	Y = 46.8460x + 11.6360	0.9999	0.99	2.98
DMF	2.76–110.51	Y = 95.4880x + 24.4180	0.9995	0.52	1.48
TMF	3.22–128.74	Y = 33.4950x + 3.5629	0.9999	1.44	4.40

Y, peak area (mAU); x, concentration of analyte (μg/mL).

**Table 4 molecules-27-04162-t004:** Accuracy and precision.

Methoxyflavones	Accuracy, *n* = 5 (% Recovery)	Precision, *n* = 5		
		Concentration (μg/mL)	Intra-Day Precision (% RSD)	Inter-Day Precision (% RSD)
PMF	109.00 ± 0.40–109.86 ± 3.72	2.61–104.50	0.06–0.29	0.34–1.70
DMF	92.23 ± 0.61–113.48 ± 0.51	2.76–110.51	0.06–0.46	1.17–3.15
TMF	92.41 ± 0.50–100.75 ± 0.88	3.22–128.74	0.03–0.95	0.52–1.83

**Table 5 molecules-27-04162-t005:** Extraction yields and total methoxyflavone contents from KP rhizomes using Plackett–Burman design ^#^.

Run	Independent Variables					Dependent Variables (Responses)	
	x_1_	x_2_ (% *v*/*v*)	x_3_ (min)	x_4_ (°C)	x_5_ (mL/g)	Extraction Yields (%)	Total Methoxyflavone Contents (mg/g Extract)
1	Ethanol	50	5	30	10	7.81	88.94
2	Methanol	50	30	70	50	16.64	104.19
3	Ethanol	95	5	70	50	6.49	255.38
4	Methanol	95	5	30	10	3.67	273.73
5	Methanol	95	30	30	10	7.70	286.44
6	Methanol	50	30	70	10	10.36	103.69
7	Ethanol	50	30	30	50	14.72	111.63
8	Ethanol	95	30	30	50	9.08	273.87
9	Methanol	50	5	30	50	10.22	83.56
10	Ethanol	50	5	70	10	9.27	95.73
11	Methanol	95	5	70	50	8.66	240.29
12	Ethanol	95	30	70	10	8.89	277.09

^#^ Results were performed using Design Expert software: x_1_, type of solvent; x_2_, organic solvent concentration; x_3_, extraction time; x_4_, extraction temperature; x_5_, solvent-to-solid ratio.

**Table 6 molecules-27-04162-t006:** ANOVA for the responses obtained from PBD screening.

Source	Sum of Squares	df	Mean Square	F-Value	*p*-Value	Significant	Remarks
Y_1_ (Extraction yields)							
Model	119.35	5	23.87	13.69	0.0031	Yes	
x_1_	0.08	1	0.08	0.05	0.8365	No	
x_2_	50.05	1	50.05	28.70	0.0017	Yes	
x_3_	37.71	1	37.71	21.62	0.0035	Yes	
x_4_	4.22	1	4.22	2.42	0.1710	No	
x_5_	27.30	1	27.30	15.65	0.0075	Yes	
Residual	10.46	6	1.74				R^2^ = 0.9194
Cor total	129.82	11					R^2^ (adj.) = 0.8522
Y_2_ (Total methoxyflavone contents)							
Model	88,146.13	5	17,629.23	293.68	<0.0001	Yes	
x_1_	9.61	1	9.61	0.16	0.7030	No	
x_2_	86,537.84	1	86,537.84	1441.61	<0.0001	Yes	
x_3_	1185.45	1	1185.45	19.75	0.0044	Yes	
x_4_	145.41	1	145.41	2.42	0.1706	No	
x_5_	267.81	1	267.81	4.46	0.0791	No	
Residual	360.17	6	60.03				R^2^ = 0.9259
Cor total	88,506.30	11					R^2^ (adj.) = 0.9125

**Table 7 molecules-27-04162-t007:** BBD experimental combinations and their results of the extraction yields and total methoxyflavone contents ^#^.

Run	Independent Variables			Dependent Variables (Responses)	
	x_2_ (% *v*/*v*)	x_3_ (min)	x_5_ (mL/g)	Y_1_ (Extraction Yields, %)	Y_2_ (Total Methoxyflavone Contents, mg/g Extract)
1	95	50	17.5	3.85	320.56
2	72.5	30	17.5	9.58	165.13
3	72.5	30	17.5	9.51	170.37
4	50	30	5	11.86	123.29
5	72.5	50	30	13.95	173.40
6	72.5	30	17.5	10.92	166.89
7	72.5	50	5	6.85	185.50
8	95	10	17.5	3.81	281.98
9	50	10	17.5	11.16	127.84
10	72.5	10	30	9.21	195.92
11	95	30	30	4.51	308.03
12	50	50	17.5	15.36	121.80
13	72.5	10	5	5.08	189.75
14	50	30	30	15.06	138.89
15	72.5	30	17.5	9.86	155.02
16	95	30	5	2.68	321.05
17	72.5	30	17.5	9.86	159.19

^#^ Results were performed using Design Expert software: x_2_, ethanol concentration; x_3_, extraction time; x_5_, solvent-to-solid ratio.

**Table 8 molecules-27-04162-t008:** Analysis of variance for the fitted polynomial models after eliminating non-significant terms.

Source	Sum of Squares	df	Mean Square	F-Value	*p*-Value	Significant	Remarks
Y_1_ (Extraction yields)							
Model	233.60	3	77.87	50.01	<0.0001	Yes	R^2^ = 0.9203
x_2_	186.13	1	186.13	119.53	<0.0001	Yes	R^2^ (adj.) = 0.9019
x_3_	14.45	1	14.45	9.28	0.0094	Yes	Adeq. precision = 22.65 % CV = 9.85
x_5_	33.02	1	33.02	21.20	0.0005	Yes	
Residual	20.24	13	1.56				
Lack of fit	5.55	3	1.85	6.47	0.0566	No	
Pure error	1.30	4	0.33				
Y_2_ (Total methoxyflavone contents)							
Model	74,317.50	3	18,579.37	139.70	<0.0001	Yes	R^2^ = 0.9790
x_2_	64,770.84	1	64,770.84	487.02	<0.0001	Yes	R^2^ (adj.) = 0.9720
x_2_^2^	8007.07	1	8007.07	60.21	<0.0001	Yes	Adeq. precision = 31.50 % CV = 5.93
x_3_^2^	1168.82	1	1168.82	8.79	0.0118	Yes	
Residual	1595.92	13	132.99				
Lack of fit	1444.30	9	180.54	4.76	0.0743	No	
Pure error	151.62	4	37.91				

**Table 9 molecules-27-04162-t009:** Verification of the experimental results.

Run	Total Yields (%)			Total Methoxyflavone Contents (mg/g Extract)		
	Predictive Value	Experimental Value	Error (%)	Predictive Value	Experimental Value	Error (%)
1	16.95	16.33	3.66	327.25	322.48	1.48
2	16.95	16.19	4.48	327.25	323.62	1.12
3	16.95	16.53	2.48	327.25	315.61	3.69
4	16.95	16.15	4.71	327.25	331.69	1.34
5	16.95	16.56	2.30	327.25	321.81	1.69

## Data Availability

The data presented in this study are contained within the article.

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
