# Peer review of "Optimization of Ultrasound-Assisted Extraction of Yields and Total Methoxyflavone Contents from Kaempferia parviflora Rhizomes"

_molecules, 2022, doi:10.3390/molecules27134162_

Round 1

Reviewer 1 Report

The manuscript "Optimization of Ultrasound-assisted Extraction of Yields and Total methoxyflavone Contents from Kaempferia parviflora Rhizomes" is devoted to studying the extraction of 3,5,7,3',4'-pentamethoxyflavone, 5,7-dimethoxyflavone, and 5,7,4'-trimethoxyflavone from  Kaempferia parviflora rhizomes using  ultrasound-assisted extraction with ethanol-water mixtures. Plackett-Burman design and Box-Behnken design were used to optimize extraction condition. HPLC-DAD was used for analysis. Analytical method is described in detail. The results of this work may be useful for expanding the use of plant material as a basis for pharmaceutical products.

The manuscript is well structured, and methods of extraction and analysis are described in detail. This manuscript can be published in the Molecules journal after minor revision taking into account some of the remarks described below 

1.      Section 2.7. Experimental design: The authors said “Prior to conducting the PBD experiments, the high and low levels of each variable were picked out with respect to the preliminary studies.” It would be better to specify the results of the preliminary studies. Also, the choice of parameters for the Plackett-Burman design should be described clearer. In my opinion, there are a lot of parameters for 2 levels… It would be better to use another DOE.

Reviewer 2 Report

The scientific content of the manuscript is informative. 

Major points:

1. The authors should provide the stacked chromatogram containing the crude extract and the standards.

2. The HPLC chromatograms of methanol extract should be provided to indicate the difference between the methanol and ethanol extracts. The ratio of PMF/DMF/TMF in both processes should be clarified. The authors also discuss why the solvents were used based on previous reports. A comparison between the UAE and other common extracts should be provided in the Discussion part.

3. Quantitative HPLC method should be provided in Section 2.6 with more information to determine the content of flavones.

4. According to the results shown in the manuscript, the extraction using methanol gave better data in both yield and methoxyflavone content. However, the authors said that the effect of solvent type was insignificant. Please clarify. The authors need to provide a discussion about this.

4. Please rewrite the Conclusion part with the main results of the manuscript. 

Minor points: 

- 2.4. Please revise "By diluting each ...., different concentrations of DMF, PMF, and TMF were obtained".

- Delete the names of flavones in footnote of Tables 3 and 4.

Reviewer 3 Report

I recommend the reviewed manuscript Manuscript entitled: „ Optimization of Ultrasound-assisted Extraction of Yields and Total methoxyflavone Contents from Kaempferia parviflora Rhizomes“ to be published in Molecules. 

The manuscript focused on extraction of the main flavones from Kaempferia parviflora.

Experimental part is well set, experimental design consist of two parts, screening of the extraction parameters and then optimization of the selected crucial variables. Model validation is well explained.

Additionally, HPLC-DAD method was developed for determining flavones (PMF, DMF, and TMF) in KP extracts. Method is fully validated and all validation data are presented. The developed HPLC-DAD method was found to be acceptable for determining PMF, DMF, and TMF in KP extracts obtained from the UAE optimization without the need for sample preparation.

Round 2

Reviewer 2 Report

The authors have revised the manuscript thorougly.